# Methods of Constructing Time Series for Predicting Local Time Scales by Means of a GMDH-Type Neural Network

Łukasz Sobolewski * and Wiesław Miczulski 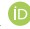

Institute of Metrology, Electronic and Computer Science, University of Zielona Gora, 65-417 Zielona Gora, Poland; w.miczulski@imei.uz.zgora.pl
* Correspondence: l.sobolewski@imei.uz.zgora.pl; Tel.: +48-68-328-2274

**Abstract:** Ensuring the best possible stability of UTC(k) (local time scale) and its compliance with the UTC scale (Universal Coordinated Time) forces predicting the [UTC-UTC(k)] deviations, the article presents the results of work on two methods of constructing time series (TS) for a neural network (NN), increasing the accuracy of UTC(k) prediction. In the first method, two prepared TSs are based on the deviations determined according to the UTC scale with a 5-day interval. In order to improve the accuracy of predicting the deviations, the PCHIP interpolating function is used in subsequent TSs, obtaining TS elements with a 1-day interval. A limitation in the improvement of prediction accuracy for these TS has been a too large prediction horizon. The introduction in 2012 of the additional UTC Rapid scale by BIPM makes it possible to shorten the prediction horizon, and the building of two TSs has been proposed according to the second method. Each of them consists of two subsets. The first subset is based on deviations determined according to the UTC scale, the second on the UTC Rapid scale. The research of the proposed TS in the field of predicting deviations for the Polish Timescale by means of GMDH-type NN shows that the best accuracy of predicting the deviations has been achieved for TS built according to the second method.

**Keywords:** timescale predicting; time series analysis; GMDH type neural network; data preparation

## 1. Introduction

The International Bureau of Weights and Measures (BIPM), based on strictly defined algorithms [1], determines the UTC scale (Universal Coordinated Time) on the basis of previously properly prepared and verified measurement data from local and remote comparisons of over 700 atomic clocks [2]. These data are sent to BIPM by the National Metrological Institutes (NMI), Designated Institutes (DI), and other laboratories responsible for the implementation of local time scales UTC(k). UTC(k) scales are the basis for determining the official time in individual countries. They are implemented on the basis of commercial cesium atomic clocks or hydrogen masers, enabling the determination of these scales with an uncertainty of, respectively $10^{-12}$ and $10^{-15}$.

Once a month, BIPM publishes the [UTC-UTC(k)] deviations for the previous month in the "Circular T" bulletin with a five-day interval, which indicate the discrepancy of the UTC(k) time scales in relation to UTC. They are designated as daily average values for MJD (Modified Julian Date) days, that end with digits 4 and 9. Ensuring compliance of UTC(k) with UTC at the level of discrepancy not exceeding $\pm 10$ ns guarantees the stability of this scale on a level not worse than 1 s per 274,000 years and puts it in the best group of local time scales in the world. It is of great importance in the economy, research, and defense. The problem of maintaining the best possible compliance of UTC(k) with UTC is due to the delay (8 to 12 days) in the next month's publication on the BIPM ftp server of the "Circular T" bulletin containing the [UTC-UTC(k)] deviations for each UTC(k) time scale. This delay is due to the complexity and time-consuming nature of the UTC determination process.

Ensuring the best possible stability of UTC(k) and its compliance with UTC forces the UTC(k) generation system to make appropriate adjustments to the output signal of the

atomic clock. In view of the delay in publishing the "Circular T" bulletin, this can only be resolved by predicting the deviations. The literature on predicting the deviations for UTC(k) present only four analytical methods based on stochastic differential equations [3], Allan deviations [4], Kalman filter [5], and linear regression [6]. Based on their own research, NMIs and DIs decided individually to choose one of the four methods listed for predicting deviations for UTC(k). The basis for predicting deviations for UTC(k) in a given month are input data, prepared for each of the methods, based on [UTC-UTC(k)] deviations, published by BIPM for the previous month. Polish Timescale UTC(PL) is determined by the Central Office of Measures (GUM—polish for Główny Urząd Miar). For predicting [UTC-UTC(PL)] deviations, a linear regression method had been used. This method, similar to the other three mentioned methods, is a time-consuming method that must be carried out by a person with extensive metrological experience and knowledge of the operation of an atomic clock and devices cooperating with it. The quality of predicting the deviations is determined by many factors, including the quality of the historical data being the basis for predicting, the instability of the applied atomic clock, the stability of the clock power source, and the climatic conditions. In 2006, the Institute of Metrology, Electronics and Computer Science of the University of Zielona Góra (IMEI) started cooperation with GUM in the field of improving the quality of predicting the deviations. The aim of this cooperation is to achieve prediction errors not exceeding the required value of $\pm 10$ ns for UTC(k) timescales belonging to the first group.

The analysis of the work carried out so far at the GUM on the timescale prediction has shown that the historical data of UTC(PL) include classified knowledge that can be extracted by a neural network (NN). The proposal of applying NN for predicting deviations for UTC(k) resulted from its property, consisting of the possibility of building models by an inductive method. NNs are very suitable tools used to solve problems of a nonlinear nature [7]. The application of the NN for predicting deviations requires appropriate preparation of input data [8].

In the first half of 2011, paper [9] presents the results of research on the possibility of innovative application of MLP, RBF and GMDH-type NNs for predicting UTC(k). According to the authors' knowledge, it is the first publication in the world presenting the initial results of research on the possibility of applying NNs for predicting deviations for UTC(k). These studies show that a GMDH-type NN increases the accuracy and simplifies the process of predicting deviations compared to the MLP- and RBF-type NNs and the linear regression method used so far by GUM. In subsequent publications of the authors, including [10–14], the results of work on improving the process of data preparation and UTC(k) prediction by means of the NN have been presented.

For several years, a few articles have been published on the application of NNs for predicting deviations for time scales in relation to the UTC scale. The paper [15] presents the first results of research on the application of a one-dimensional convolutional neural network (1D-CNN) for predicting the Japanese UTC(NMIJ) timescale. They also indicate an improvement in the quality of prediction in relation to the method based on the Kalman filter, which is still used there. On the other hand, work [16] presents the results of research on an integrated model consisting of a gray GM (1,1) and NN model for UT1-UTC prediction. These predictions are essential for space navigation applications and for the precise determination of the orbits of artificial Earth satellites.

The papers published in recent years with the application of NNs for predicting UTC(k) have been the motivation for the development of the article, the aim of which is to present the results of the authors' work on the methods of constructing time series for NNs, increasing the accuracy of predicting the deviations for UTC(k). The order of introducing the new developed time series is closely related to the use of the UTC scale, and from 2012 to the UTC and UTC Rapid scales [17]. Chapter 2 of the article provides a short characteristic of the GMDH-type NN and the rationale for its selection for predicting the deviations for UTC(k) scale. Chapter 3 presents the construction of five UTC-based time series. The method of constructing these time series has been assessed on the example

of predicting the UTC(PL) scale based on the GMDH-type NN. The next chapter presents the construction of two time series based on the UTC and UTC Rapid scales. The method of constructing these time series has been assessed in the same way as in Chapter 3. Based on the statistical analysis of the obtained prediction results, it is indicated that the TS4_1 time series is the best for predicting deviations for UTC(k). However, in the event of failure or replacement of the atomic clock, the TS5_1 time series should be used. Both time series are based on the UTC and UTC Rapid scales. Since October 2016, they have been used to predict deviations for UTC(PL) implemented by the cesium atomic clock, and since June 2018 by the hydrogen maser.

## 2. GMDH-Type Neural Network

The main problem in designing NNs is the selection of the optimal network structure in such a way that it gives the best results. The research results, which are presented in [18] have shown good prognostic properties of MLP and RBF NNs for predicting the deviations for UTC(PL). However, an important problem in this case has been the proper selection of the NN structure, which has affected the final accuracy of deviation prediction enabling correction of the atomic clock gait. These activities have been very time-consuming (for MLP type NN it takes about 50 h, and for RBF type NN about 12 h) and do not guarantee the optimal selection of the NN structure.

The importance in solving a problem using an NN is to train it, i.e., select parameters (weights) of neurons, so that the output values from the network match those of the training pattern. Therefore, it is necessary to search for new solutions in the field of modeling algorithms based on the processing of empirical data [19]. Such a solution may be the GMDH (Group Method of Data Handling) method belonging to the class of evolutionary algorithms [9]. This method is used in many fields related mainly to data acquisition, prediction, system modeling, and optimization. This method was proposed in 1968 by the Combined Control Systems group from the Institute of Cybernetics in Kiev, headed by Prof. A. G. Ivakhnenko [9]. GMDH algorithms automatically find correlations in data. The GMDH idea can be successfully applied in the design of NNs.

The application of the GMDH method at the stage of designing the NN allows us to overcome the main problem mentioned in the first paragraph of this chapter. The structure of the GMDH network is created automatically, only on the basis of prepared training and testing data sets. That is why skillful preparation of these data sets is so important. During the training process, the network develops and evolves as long as it leads to the improvement of its effectiveness [9,20]. Before the next layer of neurons is added to the current network structure, the components of the new layer are subjected to selection for precision of processing. Neurons that do not meet the criterion for assessing the condition imposed, i.e., the processing error associated with these neurons is too large, are eliminated from the structure of the network.

The advantages of the GMDH-type NN presented above have determined its application in the prediction of deviations for UTC(PL). Predicting the deviations for the Polish Timescale UTC(PL) by means of GMDH-type NN has been carried out on the basis of the method of time series analysis by means of the commercial GMDH Shell tool. Figure 1 presents the GMDH Shell application solver view, in which the values of the NN training parameters are selected. A detailed description of the selection of these parameters is presented in [21]. The duration of the entire process of data preparation, training the GMDH-type NN, and receiving the prediction result is approximately 50 min.

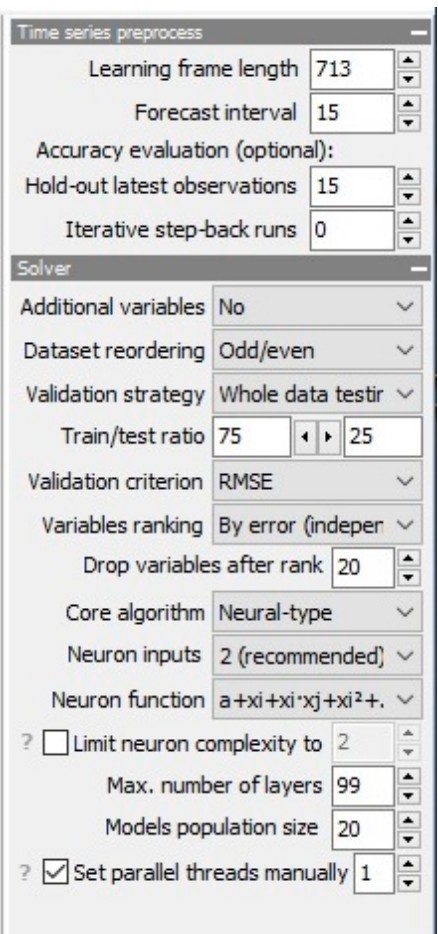

**Figure 1.** GMDH Shell solver application view.

### 3. Construction of Time Series Based on the UTC Scale

*3.1. The Method of Constructing the TS1_5 and TS2_5 Time Series Prepared with the 5-Day Interval*

An important problem of applying NNs for predicting the [UTC-UTC(k)] deviations is the proper preparation of training and testing data sets. Compared to the applied analytical models [3–6], NNs for predicting deviations require a much larger number of time series elements [22] in the training and testing process to achieve the best possible quality of predicting the [UTC-UTC(k)] deviations. The paper [23] presents the results of the research that determined the minimum number of time series elements, ensuring the required quality of UTC(PL) prediction.

Predicting the values of [UTC-UTC(k)] deviations by means of NNs for the selected UTC(k) scale is based on a time series $x(t)$, the values of which (Figure 2) are determined at times $t_0$ to $t_n$, where $t_0$ is the day of the first known value of the time series, and $t_n$ is the day of the last known value of the time series in the month preceding the prediction. The values of the time series elements for the last month preceding the prediction designation may be determined not earlier than on day $t_{pub}$. This is the date when BIPM publishes the [UTC-UTC(k)] deviation values for previous month. They are designated for MJD days that end with digits 4 and 9, i.e., with a 5-day interval ($t_n - t_{n-1} = 5$).

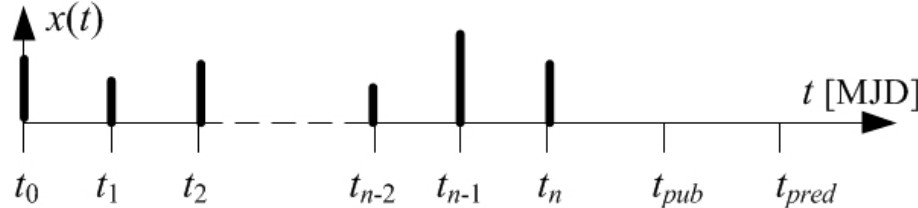

**Figure 2.** Illustration of time series $x(t)$.

The time series $x(t)$ characterizes the time instability of the clock implementing UTC(k) with respect to UTC. Its values are calculated based on the relation

$$x(t) = xa(t) + xb(t) = UTC(t) - clock_k(t),\tag{1}$$

where

$xa(t)$—historical results of phase time measurements between 1 pps (pulse per second) signals from UTC(k) and the clock realizing UTC(k), determined with a one-day interval according to the relationship:

$$xa(t) = UTC_k(t) - clock_k(t),\tag{2}$$

$xb(t)$—values of [UTC-UTC(k)] deviations determined by BIPM for UTC(k) in relation to UTC, which are designated as MJD days that end in digits 4 and 9 according to the formula

$$xb(t) = UTC(t) - UTC_k(t).\tag{3}$$

The time $t_{pred}$ is the day when the deviation for UTC(k) is predicted by the GMDH-type NN. Striving to achieve the highest possible accuracy of UTC(k) prediction also forces the prediction process to keep the prediction horizon as low as possible, i.e., the time interval $t_{pred} - t_n$, which depends on the publication date ($t_{pub}$) of the $xb(t)$ deviation values. Taking into account the time $t_{pub}$ and the need to make a prediction of $xb_{pred}(t_{pred})$ deviation for the next MJD day that ends with digits 4 or 9, the prediction of deviation is carried out between the 10th and 20th day of the following month. Establishing predictions for the following days (4 and 9) and in the following months for the same training data will cause, regardless of the analytical and NN methods used for prediction, a significant increase in prediction errors. Practically, it means that one value of the predicted $xb_{pred}(t_{pred})$ deviation is determined in a given month. However, to determine the prediction in the next month, it is necessary to supplement the $x(t)$ time series with a new group of data $i + 1$ (Figure 3), taking into account the new values of $xb(t)$ deviations from the month preceding the prediction, published by BIPM. Such action means the necessity to re-run the full process of determining the new prediction value for the next month.

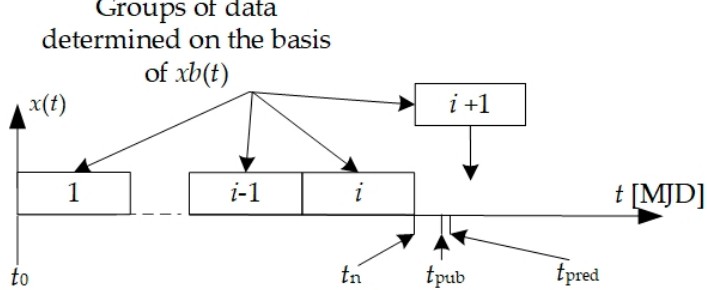

**Figure 3.** A method of constructing a time series based on the $xb(t)$ values published by BIPM.

The basis for the preparation of training data for the NN in the form of the TS1_5 time series is the relationship (1). These data are determined with a 5-day interval. The TS1_5

time series is the first group of data on the basis of which GMDH-type NN training and prediction of $x_{pred}$ values has been performed. In the next step the determination of the $xb_{pred}(t_{pred})$ prediction deviation value for UTC(k) based on the relationship

$$xb_{pred}(t) = x_{pred}(t_{pred}) - xa(t_{pred}), \tag{4}$$

has been carried out.

For a cesium clock in the TS1_5 time series, a linear trend component $x_r(t)$ and a variable component $x_d(t)$ are included (Figure 4).

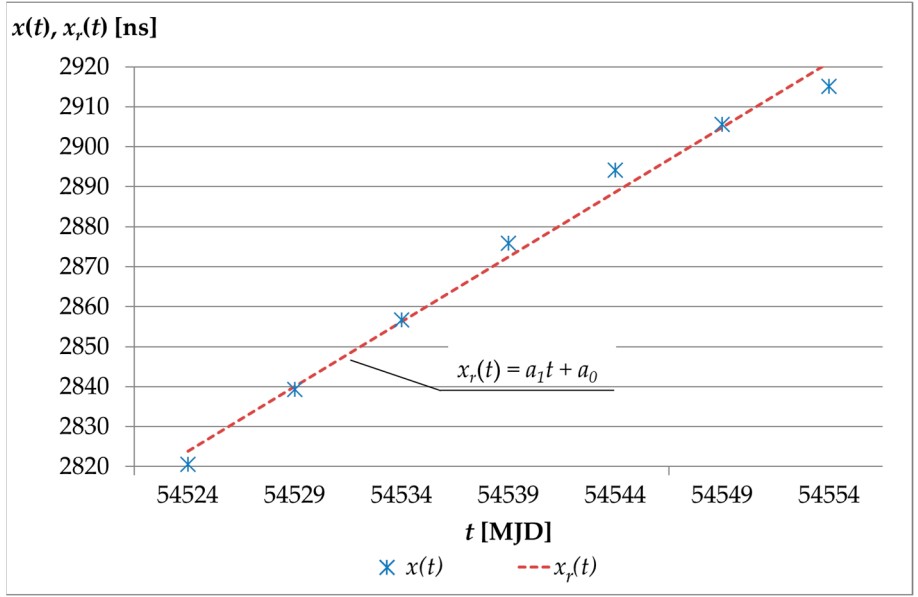

**Figure 4.** Sample set of the phase time $x(t)$ and the trend $x_r(t)$ for a one-month period for UTC(PL).

Small values of $x_d(t)$ deviations from the $x_r(t)$ trend may cause the NN to adopt the trend as important information in the training process. This, in turn, may deteriorate the results of predicting the $xb_{pred}$ deviations for UTC(k). This determined the preparation of the second time series TS2_5. Its final form was obtained as a result of the decomposition of the TS1_5 time series, from which the long-term trend of changes in the phase time $x_r(t)$, described by the linear regression equation, has been eliminated. The individual elements of the TS2_5 time series, constituting the values of deviations from the trend, have been calculated from the relation

$$x_d(t) = x(t) - x_r(t). \tag{5}$$

The introduction of data defined by the TS2_5 time series at the input of the NN will cause the adoption of a model describing deviations from the trend by the NN in the training process. The predicted values of the $x_{pred}$ deviations have been calculated from the relation (4) after taking into account the determination of $x_{pred}(t)$ from the relation

$$x_{pred}(t) = x_{rpred}(t_{pred}) - x_{dpred}(t_{pred}). \tag{6}$$

*3.2. The Method of Constructing the TS1_1, TS2_1, and TS3_1 Time Series Prepared with the 1-Day Interval*

Predicting UTC(k) by means of the NN and the time series TS1_5 and TS2_5 requires the necessary number of n elements of the time series, covering too long a period of time. This time period is determined by the five-day time interval between the $xb(t)$ deviations determined by the BIPM for each UTC(k). Given the $xb(t)$ values determined with a one-day interval, the prepared time series would further contain the necessary number of n elements, but would include a five times shorter time interval in relation to the TS1_5

and TS2_5. Therefore, it has been proposed to use the PCHIP (Piecewise Cubic Hermite Interpolating Polynomial) interpolation function, available in the MATLAB tool for the $xb(t)$ data set. The mathematical model determined on its basis makes it possible to extend the set of $xb(t)$ deviations by calculating their values for each day of the analyzed period of time. Based on the simulation work presented in [21], the Hermite interpolation turned out to be several times better than the linear interpolation and the cubic spline functions.

The properties of the interpolating function also show that the obtained values of $xb(t)$ in the interpolation nodes falling on MJD days that end with digits 4 and 9 are exactly equal to the values of $xb(t)$ determined by BIPM (Figure 5). This is important because the prediction values for these days are the basis for correcting UTC(k).

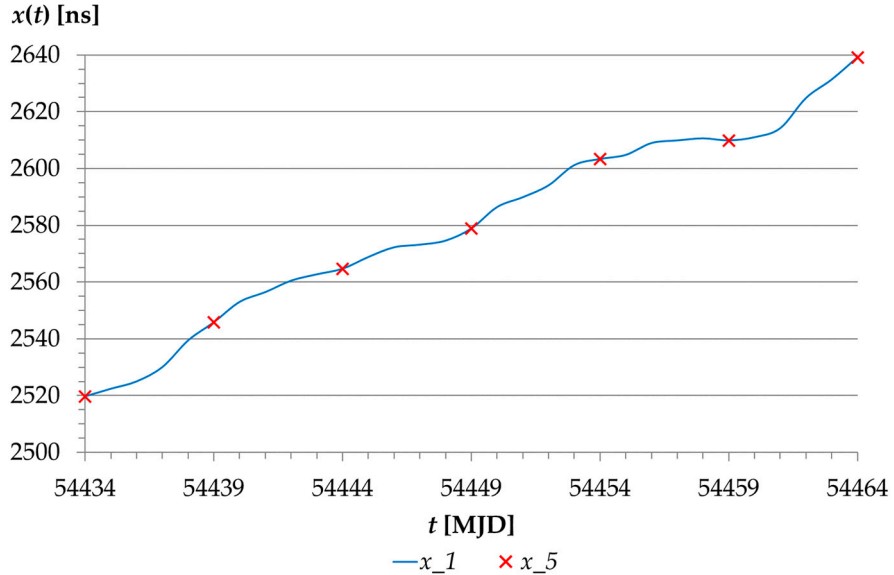

**Figure 5.** Exemplary values of $x(t)$ calculated from (1) for $xb(t)$ determined with the 1-day ($x\_1$) and 5-day ($x\_5$) interval for a period of one month for UTC(PL).

The values of $x(t)$ elements for the proposed new time series TS1_1 and TS2_1 are determined with a 1-day interval according to the described rule for the TS1_5 and TS2_5 (Section 3.1). The advantage of such preparation of the TS1_1 and TS2_1 time series is also that in the training process of the NN, all changes in the gait of the clock occurring between MJD days that end with digits 4 and 9 (Figure 5), which affect the prediction result, are taken into account.

In certain situations, e.g., failure, it may be necessary to replace the atomic clock implementing the UTC(k) national scale with another clock. Then there may be a lack of time series elements to correctly perform the UTC(k) deviations prediction. This fact resulted in the creation of another time series (TS3_1), containing elements with the 1-day interval. The values of these elements are only the $xb(t)$ deviation values, published by BIPM, interpolated by the PCHIP function.

### 3.3. Assessment of the Construction Method of the Proposed Time Series Based on the UTC Scale

The method of constructing the proposed TS1_5, TS2_5, TS1_1, TS2_1, and TS3_1 time series has been assessed on the example of predicting the UTC(PL) scale by means of the GMDH-type NN. Prediction has been carried out for all time series in the period of five months, from January to May 2008 (from MJD 54479 to MJD 54599) for days that end with the digits 4 or 9. The prediction horizon is from 10 to 20 days. Input data for the NN for all time series are available from 4 January 2006.

The basis for the evaluation of all proposed time series are the residual values, determining the discrepancy between the predicted $xb_p(t_{pred})$ deviation and $xb(t_{pred})$ deviation published by BIPM on the same prediction day ($t_{pred}$), which are calculated from the relation

$$r(t_{pred}) = xb(t_{pred}) - xb_p(t_{pred}), \tag{7}$$

and selected measures of prediction quality determined on their basis [24,25].

The calculated values of the residuals for the five analyzed time series are presented in Figure 6. The values of the residuals for each time series are included in the following intervals:

- from −5.05 ns to 6.95 ns for TS1_5,
- from −4.63 ns to 7.74 ns for TS2_5,
- from −0.60 ns to 3.08 ns for TS1_1,
- from −0.07 ns to 0.40 ns for TS2_1,
- from −15.91 ns to 6.89 ns for TS3_1.

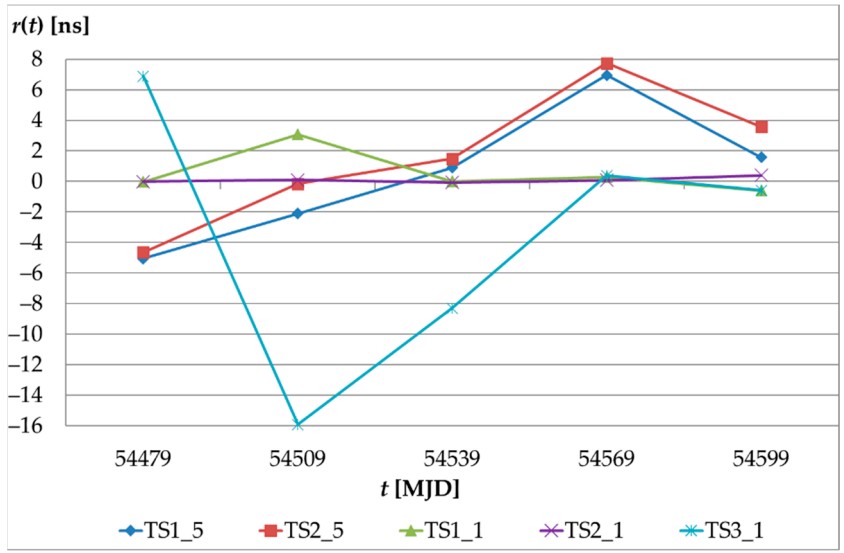

**Figure 6.** Obtained *r* residuals values for predicting $xb_p(t)$ for the TS1_5, TS2_5, TS1_1, TS2_1, TS3_1.

Table 1 presents selected measures of prediction quality [24,25]: mean error (ME), absolute mean error (MAE), mean square error (MSE) with its components (MSE$_1$, MSE$_2$, MSE$_3$) and the root of mean square error (RMSE).

**Table 1.** Obtained values of selected quality prediction measures for the TS1_5, TS2_5, TS1_1, TS2_1, and TS3_1.

| Measure of Quality of Prediction | TS1_5 | TS2_5 | TS1_1 | TS2_1 | TS3_1 |
|:---:|:---:|:---:|:---:|:---:|:---:|
| *ME* [ns] | 0.45 | 1.61 | 0.55 | 0.1 | −3.5 |
| *MAE* [ns] | 3.32 | 3.52 | 0.8 | 0.13 | 6.41 |
| *MSE* [ns2] | 16 | 19 | 2.0 | 0.04 | 74 |
| *MSE*$_1$ [ns2] | 0.21 | 2.56 | 0.31 | 0.01 | 12.8 |
| *MSE*$_2$ [ns2] | 5.15 | 6.73 | 0.01 | 0.01 | 2.25 |
| *MSE*$_3$ [ns2] | 10.9 | 9.97 | 1.67 | 0.02 | 59.4 |
| *RMSE* [ns] | 4.04 | 4.39 | 1.41 | 0.19 | 8.6 |

Based on the presented results (Figure 6) and the calculated values of prediction quality measures (Table 1), it can be concluded that:

(1) The smallest residual values have been obtained for the TS1_1 and TS2_1. This is due to the application of the PCHIP interpolation function, which makes it possible to prepare data for the GMDH-type NN with a one-day interval.

(2) Comparison of ME, MAE, and $MSE_1$ error values for all time series shows that for the TS3_1 time series the predictions are the most loaded.

(3) The deviations predictions determined for TS1_1 and TS2_1 are characterized by small values of the $MSE_2$ error component. This allows a good prediction of the variability of the predicted values in relation to the variability of the values observed for both time series, which results from the density of training data in a given month, thanks to application of the PCHIP function.

(4) In the case of results obtained for TS2_1 time series, the smallest value of the $MSE_3$ error component has been obtained in relation to the other time series. This is due to the decomposition of the TS1_1 time series, which resulted in a better compliance of the direction of changes in the prediction and the direction of changes in the predicted value for the TS2_1 time series.

(5) The comparison of the obtained values of residuals and the values of all the prediction quality measures indicates, that the best quality of predicting the deviations has been obtained using the TS1_1 and TS2_1 time series. This is due to the application of $xb(t)$ values determined with a one-day interval, thanks to the application of the PCHIP function. The research results for these time series for a longer period of time presented in [10,24] confirm that the TS1_1 and TS2_1 enable the best quality for predicting deviations. However, these studies show that the TS1_1 is more suitable for practical application. The TS2_1 is characterized by a slightly worse quality of prediction and is more time-consuming to prepare.

(6) The values of the TS3_1 are determined only on the basis of the $xb(t)$ deviation values published by BIPM. The achieved unfavorable large value of $xb(t)$ for the MJD 54499 day, amounting to 26.6 ns (Figure 7), directly influenced the large residual value for MJD 54509, as well as the MSE error and its components $MSE_1$ and $MSE_3$. Such a large residual value results from a large change in UTC-UTC(PL) deviations in the period from MJD 54479 to MJD 54509. The proposed TS3_1, despite the non-optimal residual values and the determined quality measures of prediction achieved, is a good alternative for predicting $xb(t_{pred})$ in the case of absence of the $xa(t)$ values due to a failure of the atomic clock or its replacement.

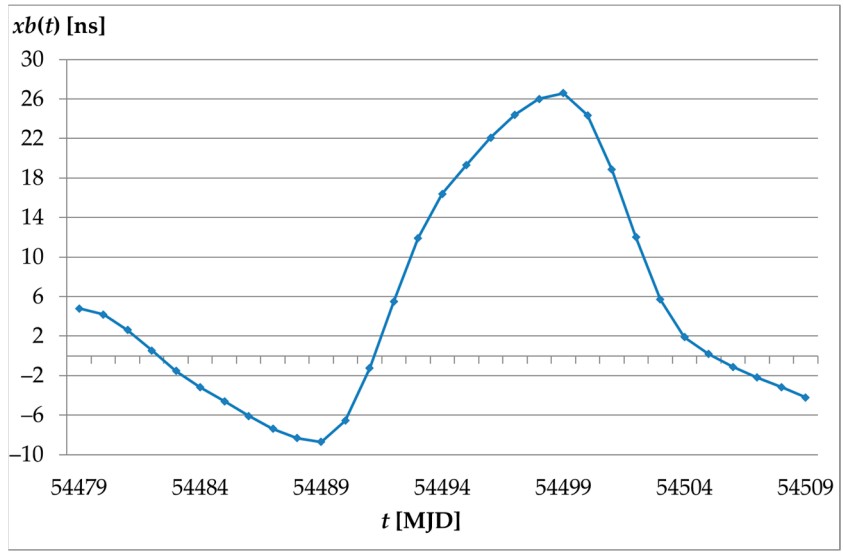

**Figure 7.** Values of $xb(t)$ deviations for a period of one month (from MJD 54479 to MJD 54509) after interpolation with the PCHIP function.

## 4. Construction of Time Series Based on the UTC and UTC Rapid Scale

### 4.1. The Method of Constructing the TS4_1 and TS5_1 Time Series Prepared with the 1-Day Interval

One of the factors influencing the quality of prediction by means of the GDMH-type NN is the prediction horizon. The delay in the publication of $xb(t)$ deviations by BIPM means that the first prediction can be made between the 10th and 20th day of the month. This has a negative impact on the result of the prediction obtained, and consequently on maintaining the best possible compliance of UTC(k) with UTC. BIPM launched "A Rapid UTC" project in 2012, the purpose of which is to accelerate the transmission of information about the divergence of UTC(k) in relation to UTC. Based on the data from clocks and time transfer systems sent daily to BIPM, an auxiliary UTC Rapid (UTCr) scale is calculated for each day every week [17]. When determining the UTCr scale, the weights of individual clocks in the current month are assigned with the values that have been determined for the UTC scale in the previous month. Based on the determined UTCr scale, the [UTCr-UTC(k)] deviations for the previous week are published every Wednesday on the BIPM ftp server. They are determined according to the relation

$$xbr(t) = UTCr(t) - UTC_k(t). \tag{8}$$

The publication by BIPM of the $xbr(t)$ values every Wednesday enables the first prediction of [UTCr-UTC(PL)] to be determined each week, for the MJD day that ends with digit 4 or 9 with a shorter prediction horizon of three to seven days, compared to the prediction horizon for time series built on the basis of the UTC scale. This makes it possible to introduce another method of constructing time series (Figure 8), the values of which are determined with a one-day interval. The basis for the preparation of these time series are the $xb(t)$ and $xbr(t)$ deviations determined on the basis of the UTC and UTCr scales, respectively.

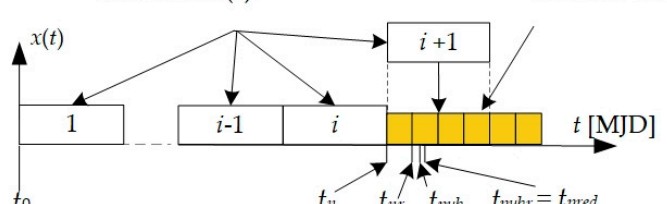

**Figure 8.** A method of constructing a time series based on the $xb(t)$ and $xbr(t)$ values published by BIPM.

The proposed time series (TS4_1 and TS5_1) consist of two subsets of elements prepared according to the principle presented in Figure 8. The first subset of the data group, from 1 to $i$ months, is determined on the basis of relation (1) for the TS4_1 and relation (3) for the TS5_1. The values of $xb(t)$ appearing in both relations are calculated on the basis of the PCHIP function. The thus determined values of the TS4_1 and TS5_1 are known from day $t_0$ to day $t_n$ for which the last $x(t_n)$ value is known before the publication day $t_{pub}$. From the properties of the PCHIP function, quoted in Section 3.2, it follows that the determined values of $xb(t)$ falling on MJD days that end with digits 4 and 9 are exactly equal to the values of $xb(t)$ determined by BIPM. It is very important because the values of predictions for these days are the basis for correcting UTC(k). However, the values of $xbr(t)$ are determined by BIPM on the basis of the UTC Rapid scale, and therefore for MJD days that end with the digits 4 and 9, they will not always reach the values equal to $xb(t)$.

In order to compare the values of $xb(t)$ determined on the basis of the PCHIP function and xbr(t), their differences are calculated from the relation

$$\Delta xb(t) = xb(t) - xbr(t). \tag{9}$$

Figure 9 shows an exemplary set of $\Delta xb$ values for the period of 491 days (from MJD 56204 to MJD 56694). In the analyzed period of time, there are 99 MJD days that end with the digits 4 and 9, but only for five MJD days (56274, 56499, 56544, 56559, and 56659) does $\Delta xb = 0$. However, for all 491 days, the values $\Delta xb = 0$ are only for 13 days. Figure 9 also shows that the discrepancies between the values of xb and xbr are within the interval of $-5.7$ ns to 6.2 ns for the entire analyzed period, and for 99 MJD days ending with the digits 4 and 9 in the interval of $-3.7$ ns to 4.0 ns.

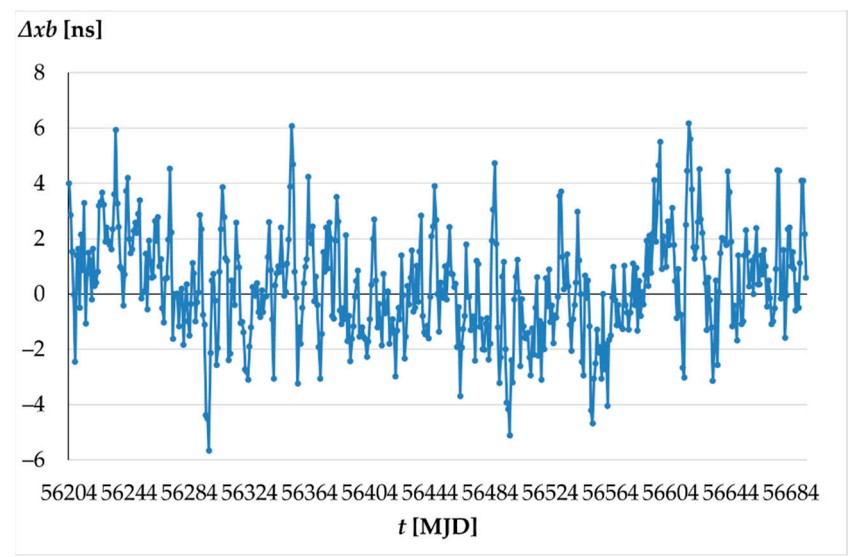

**Figure 9.** Set of $\Delta xb(t)$ values for a period of 491 days.

The determined statistical parameters for the values of $\Delta xb(t)$ for the period of 491 days ($\Delta xb_{aver} = 0.28$ ns and standard deviation $\sigma = 1.94$ ns) and 375 values of $\Delta xb$ within the range from $-1$ ns to 1 ns confirm that the adopted PCHIP function is a very good model to describe the values of $xb$.

The second subset completes the TS4_1 or TS5_1 with data groups determined on the basis of the UTCr scale (Figure 8) between days $t_n$ and $t_{nr}$ (7 days) containing elements with values determined on the basis of relation

$$x(t) = xa(t) + xbr(t) = UTCr(t) - clock_k(t), \tag{10}$$

for TS4_1 and on the basis of relation (8) for TS5_1. The values of $xbr(t)$ are published by BIPM on day $t_{pubr}$ (Figure 8), which can also be the prediction date ($t_{pred}$) ($t_{pred} \geq t_{pubr}$).

A significant advantage of the TS4_1 and TS5_1 is the weekly introduction of new data groups calculated on the basis of the relation (8) or (10) and determination of successive $xb_p(t_{pred})$ prediction values in the same month. At the time of publishing a new "Circular T" bulletin, a new data group for the next month $i + 1$ is created (Figure 8), based on the relation (1) for TS4_1 or (3) for TS5_1, which replaces the previous data for the relevant days on the basis of relation (8) or (10). It thus supplements the first data subset for the series TS4_1 or TS5_1 defined with a one-day interval.

### 4.2. Assessment of the Construction Method of the Proposed Time Series Based on the UTC and UTC Rapid Scales

The method of constructing the TS4_1 and TS5_1 series has been assessed on the example of UTC(PL) prediction based on the UTC and UTC Rapid scales by means of the GMDH-type NN. The research has been carried out over a period of 17 months, from MJD 56204 to MJD 56699, determining the first prediction in each week for days that end with digits 4 or 9. Input data for the GMDH-type NN are the TS4_1 and TS5_1 time series. In order to compare the quality of determined predictions for two methods of preparing time series in the same period of time, the TS1_1 has been selected from the first method of the time series preparation. For this time series, it is possible to calculate only one prediction per month. Figure 10 shows the results of the calculations of the residuals ($r$) for determining the $xb_p(t_{pred})$ predictions for UTC(PL) by the GMDH-type NN for the TS1_1, TS4_1, and TS5_1 time series.

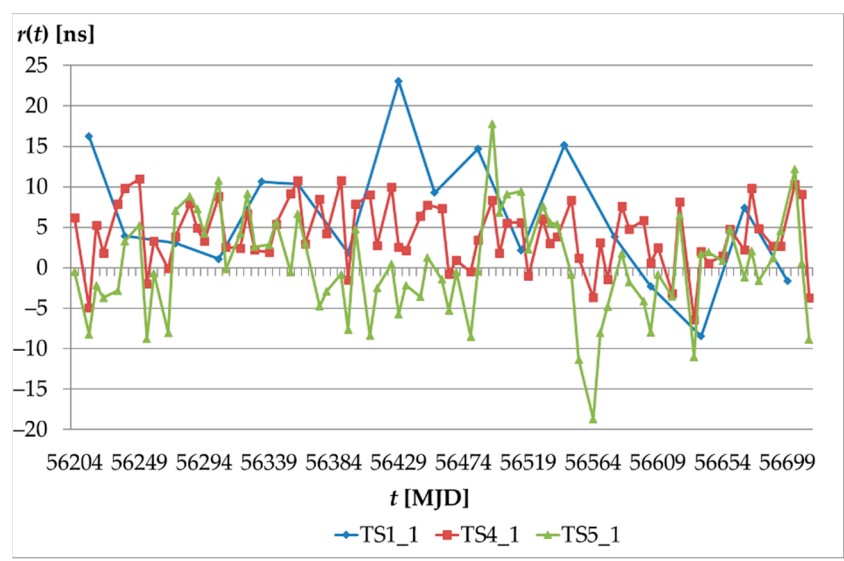

**Figure 10.** Obtained $r$ residuals values for predicting $xb_p(t)$ for the TS1_1, TS4_1, and TS5_1.

The obtained values of the residuals ($r$) for analyzed time series are included in the following intervals:

- from −8.43 ns to 23.04 ns for TS1_1,
- from −6.42 ns to 10.99 ns for TS4_1,
- from −18.67 ns to 17.82 ns for TS5_1.

On the basis of the values of the residuals ($r$) calculated from the relation (7), the prediction quality is assessed (Table 2) for the TS1_1, TS4_1, and TS5_1 using the same prediction quality measures presented in Section 3.3.

**Table 2.** Obtained values of selected quality prediction measures for the TS1_1, TS4_1, and TS5_1.

| Measure of Quality of Prediction | TS1_1 | TS4_1 | TS5_1 |
|---|---|---|---|
| $ME$ [ns] | 6.5 | 4.1 | 0.22 |
| $MAE$ [ns] | 7.9 | 4.9 | 5.0 |
| $MSE$ [ns2] | 102 | 33 | 40 |
| $MSE_1$ [ns2] | 43 | 17 | 0.04 |
| $MSE_2$ [ns2] | 5.8 | 0.01 | 1.5 |
| $MSE_3$ [ns2] | 54 | 17 | 38 |
| $RMSE$ [ns] | 11 | 5.8 | 6.3 |

Based on the presented results (Figure 10) and calculated values of prediction quality measures (Table 2), it can be concluded that:

(1)　Comparison of ME, MAE, and $MSE_1$ error values for three analyzed time series shows that for the TS5_1 time series, the predictions are the least loaded.

(2)　Predictions of the deviations determined by the GMDH-type NN for the TS4_1 time series are characterized by smaller $MSE_2$ and $MSE_3$ error values compared to the TS5_1 time series. This means better anticipation of fluctuations in the prediction variable and greater compliance of the direction of prediction changes in relation to the direction of changes in the predicted value. Hence, 64% of determined values of residuals for the TS4_1 time series are in the range of $\pm 5$ ns, and 96% of the determined residual values for this time series are in the range of $\pm 10$ ns. The remaining determined residual values (4%) are within $\pm 11$ ns.

(3)　From the group of TS1_1, TS4_1 and TS5_1 time series examined, the best prediction quality results were obtained for TS4_1 time series. This is due to the shorter prediction horizon, that has been achieved by introducing $xbr(t)$ deviations determined by BIPM into the construction of these time series.

(4)　The introduction in the second method of constructing the time series of $xbr(t)$ deviations determined by BIPM also makes it possible for TS5_1 to achieve better-quality predictions than for TS3_1

## 5. Conclusions

Predicting the [UTC-UTC(k)] deviations can be carried out by means of analytical or neural networks methods. Regardless of the method used, there is a need to properly prepare time series for prediction. Until now, these time series have been built with an interval of five days, which had a significant impact on the quality of predicting the deviations. This interval is determined by the way that BIPM publishes the [UTC-UTC(k)] deviations for each UTC(k) scale. The article proposes two methods of constructing time series, which have a significant impact on the quality of predicting the [UTC-UTC(k)] deviations. The GMDH-type neural network has been used for predicting the deviations.

In the first method of constructing time series, based on deviations determined according to the UTC scale, the improvement of the prediction quality has been obtained by building the TS1_1, TS_2_1, and TS3_1 time series with a one-day interval based on the PCHIP interpolation function. The limitation of the improvement in the quality of prediction has been the too long prediction horizon, from 10 to 20 days. The best quality of predicting the deviations has been obtained for TS1_1. On the other hand, the TS3_1, despite the lower quality of deviation predicting achieved, can be used in the case of failure of the atomic clock or its replacement.

The second proposed method of constructing time series is based on the deviations published by BIPM according to the UTC and UTC Rapid scales. Two time series, TS4_1 and TS5_1, have been proposed, each of which consists of two subsets. In the first subset of the data group for both time series, based on the values of $xb(t)$ deviations determined according to the UTC scale, the PCHIP function has been used to determine the values of these time series for each day. In the second subset of data for TS4_1 and TS5_1, $xbr(t)$ deviation values determined according to the UTC Rapid scale have been used, which significantly reduced the prediction horizon (from three to nine days). As a result, these activities make it possible to predict the deviations for UTC(PL) with a discrepancy not exceeding $\pm 10$ ns from the [UTC-UTC(PL)] deviations published by BIPM.

The presented research results have been the basis for the selection of the method of constructing the TS4_1 series to predict the deviations for UTC(PL) by means of the GMDH-type neural network. They have been carried out since October 2016. In the first period, the UTC(PL) scale was implemented by the cesium atomic clock, and since June 2018 it was implemented by the hydrogen maser. During the transitional period, after replacing the cesium clock with a hydrogen maser, the TS5_1 has been used for predicting the [UTC-UTC(PL)] deviations. For the hydrogen maser, the values of predicted [UTC-UTC(PL)] deviations for both time series do not exceed a few ns.

Currently, at the IMEI, in cooperation with the GUM, preliminary work is underway on the concept of preparing a new type of time series for predicting the Polish Time Scale UTC(PL).

The results and conclusions resulting from the presented research on the methods of preparing time series may be useful in predicting UTC(k) with analytical methods and with the use of NNs. The recent years indicate that works on the application of NNs for predicting UTC(k) are being carried out in other centers [15,16].

**Author Contributions:** Conceptualization, Ł.S. and W.M.; methodology, Ł.S. and W.M.; software, Ł.S. and W.M.; validation, Ł.S. and W.M.; formal analysis, Ł.S. and W.M.; investigation, Ł.S. and W.M.; resources, Ł.S. and W.M.; data curation, Ł.S. and W.M.; writing—original draft preparation, Ł.S. and W.M.; writing—review and editing, Ł.S. and W.M.; visualization, Ł.S. and W.M.; supervision, Ł.S. and W.M. All authors have read and agreed to the published version of the manuscript.

**Funding:** This research received no external funding.

**Institutional Review Board Statement:** Not applicable.

**Informed Consent Statement:** Not applicable.

**Data Availability Statement:** MDPI Research Data Policies.

**Conflicts of Interest:** The authors declare no conflict of interest.

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
