# Peer review of "Methods of Constructing Time Series for Predicting Local Time Scales by Means of a GMDH-Type Neural Network"

_applsci, doi:10.3390/app11125615_

Round 1

Reviewer 1 Report

Line 5: Po-land

Line 7: Affiliation 1; [email protected]

Introduction:  I miss the analysis of this method in the world. Since when has it been used, where has it all been used.

I also miss the novelty and goals of the research. A little more references from the field of research

Line 116: first reference to the image, then only the image

Figure 4, 5, 6, 8, 9: edit font size

Line 250: explain why there are such large discrepancies in TS3_1 compared to other data

Line348: 3.3 think 4.2

I miss the discussion in which the authors should state how they came up with their research, what the comparison is with similar research, and what needs to be added in the future to make the research even better.

Author Response

We thank for an in-depth review and valuable tips, remarks and comments on our article. We really believe that the paper has been substantially improved after considering all the comments and that it is much clearer to the reader.

Reviewer 2 Report

Dear Authors,

Great work. I found the review paper with sound scientific idea and robust methodology. The results would be very helpful for keeping the record of time scale among the different countries for better use of resources. 

I have a very small comment on your abstract. You mentioned the aim of the paper like "The article presents the results of work on two methods ...." Then you have described the first method, i.e. "In the first method ....". But then you never talked about the second method. So, I think there is something missing the writing. It would be nice for editor to answer on this point. 

Thank you.

Author Response

Dear Reviewer,

We thank for an in-depth review and valuable comment on our article. We really believe that the paper has been substantially improved after considering all the comments and that it is much clearer to the reader.

Reviewer 3 Report

  • The paper proposes the alternative methods for time series construction to predict the local time scales.
  • The paper is well written and organized. However, there are some issues that need to be addressed.
  • The significance and contributions of this study need to be better emphasized.
  • The related literature seems to be inadequate. The authors briefly mention other methods and related works.
  • Also, the authors need to compare the performance of the proposed methods with those currently available or existing ones.
  • On page 2, paragraph 3, it's unclear what "chapters" the authors refer to.

Author Response

Dear Reviewer,

We thank for an in-depth review and valuable tips, remarks and comments on our article. We really believe that the paper has been substantially improved after considering all the comments and that it is much clearer to the reader.
